# School Burnout after COVID-19, Prevalence and Role of Different Risk and Protective Factors in Preteen Students

**DOI:** 10.3390/children10050823

**Published:** 2023-04-30

**Authors:** Noémie Lacombe, Maryelle Hey, Verena Hofmann, Céline Pagnotta, Myriam Squillaci

**Affiliations:** Department of Special Needs Education, University of Fribourg, St-Pierre Canisius 21, 1700 Fribourg, Switzerland

**Keywords:** school burnout, stress, risk factor, protective factors, COVID-19

## Abstract

Background: Current data show an increase in stress among youth since the COVID-19 pandemic, raising the question of the measures to be put in place to limit it. Aim: The aim of this study is to measure the prevalence of burnout and the different risk and protective factors of burnout among students and to compare the mean scores obtained with those collected in a similar sample in 2014. Method: Perceived health, school burnout, and the different risk and protective factors among students were measured by self-reported questionnaires among a sample of 184 students (11–13 years old). Results: The results indicate significantly higher burnout scores than in 2014. Different variables were predictive of school burnout: 1. At the individual level, a low academic achievement in mathematics, somatic symptoms, and the expressed stress; 2. At the interpersonal level, lack of support from peers, teachers, and parents and a negative classroom climate; 3. At the pandemic level, the increased workload related to the consequences of COVID-19. The factors with the greatest impact are confidence in the future, perceived stress, parental supports, and mathematics results. Conclusions: An intervention program targeting these four factors among burned out students would be relevant to set up in order to reduce its prevalence.

## 1. Introduction

While burnout was initially measured in the adult professional world, the similarities between school and professional environments have prompted researchers to look at student burnout [1]. Indeed, research in the field has shown that the psychological manifestations of burnout in students are similar to those encountered in adults. These data are not surprising in themselves, as school is at the heart of children’s lives, with a large number of hours, tasks, demands, and cognitive, instrumental, and social expectations, resulting in many potential stressors that students must face [2]. When demands exceed available resources, stress can get out of hand and lead to burnout [3]. According to Salmela-Aro et al. [1], this syndrome manifests itself in three dimensions: emotional exhaustion, cynicism about school, and feelings of inadequacy, which are close to the three-dimensional model of burnout developed by Maslach and Jackson [4] for professionals. More specifically, emotional exhaustion manifests itself as chronic fatigue linked to an overload of work, such as excessive school demands, perceived as uncontrollable [5,6]. Cynicism is defined as a loss of interest and meaning in schooling and related activities. The student may display a detached and indifferent attitude towards school. Finally, the feeling of inadequacy refers to a lack of achievement at school, a feeling of not ‘fitting in’ in the role of a student [6,7].

Table 1 provides a summary of the reported prevalence of school burnout found in 11 studies focused on school burnout. According to these data, the average prevalence stands at 14.36%. Among authors who measure the prevalence of burnout using the SBI, Gabola et al. [8] report 14.6% burnout among students aged 14.8 years on average. Among slightly younger Swiss students (11–13 years), Lacombe and Squillaci [9]. find that 4.5% of students would experience a high level of burnout. The authors justify these low results by the fact that the sample was younger than in the reference study [1]. Salmela-Aro and Tynkkynen [10] reach similar results: between 6.9% and 10.3% among 15-year-old students and between 9.9 and 16.9% among 17–18-year-old students. The same study distinguishes between the prevalence of burnout among young people in academic and vocational streams. According to their results, burnout is more frequent in academic streams.

The Italian sample of Gabola et al. [8] contains 27.6% of burnout students. Compared to the Swiss sample of the same study (14.6%), the authors highlight the significant effect of age and nationality on the results. They hypothesize that Italian students do not have access to the same health services as Swiss students. Another hypothesis could be put forward, related to the COVID-19 pandemic. As the data for this study were collected during the year 2020 and the health situation in Italy was particularly critical at the beginning of the pandemic [16] it may have influenced the results.

Regarding studies, in which other measurement instruments than the SBI were used, prevalences relatively similar to the 10–15% stated by [1] were found in particular: 18.1% measured by the shortened version of the SBI, the Short Student Burnout Inventory (SSBI) [11] 6.9% and 12% measured by the SMBM [13] or 15.47% by the KABI [14].

In their Finnish study using the BBI-10, Virtanen et al. [12] surveyed 2485 students with an average age of 14.7 years. They report that 5.5% of their sample, or about 137 students, have a low engagement profile, coupled with high burnout. In addition, 40.6% of the students would correspond to a high engagement and low burnout profile, while 53.9% would have a medium level of engagement and burnout. The authors do not comment on this low result.

As for Cheung and Li [8], despite a moderate level of academic burnout in the overall sample, they found 32.2% (N = 389) of students in the burnout profile. These students reported high levels of exhaustion and cynicism and moderate levels of work inefficiency. Moreover, 57.8% of the sample fell into the ‘moderately engaged’ profile, while only 10% fell into the ‘high functioning’ profile. These prevalences, collected through the MBI-SS, were measured in a sample of Chinese students. The authors were alarmed by the high percentage found in their sample. For them, the fact that academic performance is a top priority in their country explains the fact that these adolescents, although exhausted, continue their studies and are therefore so represented in the burnout profile.

Thus, among these results and despite the different tools used, seven results out of eleven report prevalences between 7 and 18% of burnt-out students. Two results found lower prevalences and two studies higher prevalences.

To attempt to control the extent of the phenomenon, researchers investigate the risk factors of burnout, classifying them into three distinct categories [5,17,18]: individual, interpersonal, and organizational risk factors. Among the individual factors, gender and educational level are the most frequently studied variables. In several studies, girls have higher burnout scores than boys, especially in the dimensions of exhaustion and inadequacy [1,19]. However, other studies do not find significant differences between girls and boys [9,12,20,21]. As for the level of education, the results of studies show that students with a low level of education and/or learning difficulties are more likely to develop burnout than students who do not [22,23]. Moreover, current evidence shows negative correlations between academic achievement and school burnout [24,25].

Among the interpersonal factors, different variables have been identified in the research, among which peer, teacher, and parental support are central. Virtanen et al. [12] note that peer support is a protective factor against burnout, especially if it comes from peers with low levels of burnout [24]. In terms of teacher–student relationships, the more positive they are perceived to be, the better the performance and well-being of students [26,27]. A positively perceived school climate is a protective factor against burnout [23] whereas a school climate that is perceived negatively by students influences the severity of burnout [1].

Finally, parental support is also considered a protective factor against psychosocial health difficulties [23,28]. According to Meylan et al. [7], students who feel emotionally supported by their parents are less stressed by the demands of school and are less prone to burnout.

At the organizational level, research has identified various variables related to the organization of school life. According to Meylan et al. [20] students in academic tracks have higher burnout and exhaustion scores than students in vocational tracks. Salmela-Aro and Tynkkynen [10] and Bask and Salmela-Aro [29] also find this difference between the two study paths. Besides these individual, interpersonal, and organizational factors, other factors related to the pandemic must also be taken into account. Since 2020, the consequences of the COVID-19 on the health of children, adolescents, adults, and the elderly have brought to light new risk factors related to both physical and psychological health [30,31,32,33,34,35]. In this regard, a recent meta-analysis of 29 studies found that following the first year of COVID-19, 25.2% of young people showed elevated symptoms of depression and 20.5% elevated symptoms of anxiety [36]. The study by Tang et al. [35]) conducted among a sample of young people aged 6 to 17 years (*M* = 11.86, SD = 2.32), interviewed two months after the start of the pandemic, shows that 11.5% of young people presented clinical thresholds for anxiety, depression, and stress above the usual thresholds. Current data suggest that these figures have therefore doubled since the pandemic [36].

The present study aims to measure the burnout of 8th grade students one year after the lockdown in order to investigate the medium-term influence of the pandemic on their perceived health. The study follows on from research conducted in 2014 by Lacombe and Squillaci [9] which assessed the burnout in a sample of students with similar characteristics (age, school environment, Swiss canton). Thus, the mean scores in the three dimensions obtained in 2022 (T2) will be compared to those obtained in 2014 (T1) in order to check whether there is a difference in the perceived level of burnout. In addition, the study aims to identify risk factors related to the three dimensions of burnout.

This research answers the following three questions:Q1: What is the prevalence of school burnout among 8th grade pupils in the French-speaking part of Switzerland, in the canton of Fribourg? The hypothesis in line with the post COVID-19 student health reports [30] stipulates an increase in stress and depressive symptoms, which suggests an increase in social burnout.Q2: Are the mean scores in the three burnout dimensions different between T2 and T1? In line with studies on school burnout [9,12] among 12 to 16 year olds which report low burnout scores across their samples, the hypothesis is that even if the scores increase due to COVID, the score will remain correct when considering the whole sample.Q3: What are the personal, interpersonal, organizational, and COVID-19 pandemic-related factors that explain student burnout? According to previous research results, the following hypotheses can be formulated: (a) gender has no influence on school burnout [9,15]; (b) Low school performance increases the risk of burnout [3,9,11,37]; (c) Support from parents, friends, teachers reduces burnout [6,9,12]; d) Enjoying the classroom (perceived school climate) reduces the risk of school burnout [9].

## 2. Materials and Methods

Authorization to conduct the research was granted by the Directorate of Education and Cultural Affairs of the Canton of Fribourg (DFAC) in 2014 and in 2022. In 2014, 333 students in the canton of Fribourg were surveyed. The pupils filled in the paper questionnaires in the presence of the researcher. No personal data (first name, surname, photo, or video) was collected. For more details on the 2014 procedure, see Lacombe and Squillaci [9]. In 2022, the DFAC designated six institutions in which to conduct the research. After having contacted the schools’ management, direct contact was established with each teacher in order to organize the anonymous completion of the questionnaires. As the participants are minors, permission forms were completed by the parents.

The sample comprised 184 preteens (11 to 13 years old; *M* = 12.38), including 98 boys and 86 girls. The participants came from ten classes of 8^ème^ in the canton of Fribourg in Switzerland. In the canton of Fribourg, the school system consists of 8 years of elementary school (from 1st to 8th grade), concerning children from 4 to 12 years old. Then, schooling includes 3 years of secondary school (9th to 11th). Pupils then have the possibility of continuing their studies to obtain a high school diploma or to take up an internship in the professional world. The choice of pre-adolescents is justified by the lack of research on burnout scores in this sample. Indeed, many studies measure burnout in adolescents (13–18 years) see Table 1. However, it is crucial to measure school burnout in pre-adolescents in order to understand the timing of its appearance in the classroom. The sample included 78.3% students with a Swiss nationality (n = 144). Of the 21.7% foreign of participants, 12% (n = 22) were from other European countries, 4.9% (n = 9) from African countries, and 1.6% (n = 3) from Asian countries.

### 2.1. Materials and Procedure

Data collection took place in May 2022 approximately two years after the end of the pandemic lockdown. Two members of the research team administered the questionnaires to the students. The questionnaires were gathered in the classroom, using a tablet, in the presence of the class teacher. After explaining the purpose of the research, giving some instructions, guaranteeing anonymity, freedom of participation, and the possibility of interrupting the study at any time, the pupils answered the various questions individually. The presence of a researcher was guaranteed to answer potential questions of understanding.

The self-reported questionnaire consists of four separate parts. Section 1 collects socio-demographic data. Section 2 measures student burnout using the School Burnout Inventory. Section 3 measures school stress using the Daily Worry Scale. Section 4 measures interpersonal variables through questions on support. Finally, the fifth part measures several variables related to the perception of the COVID-19 pandemic.

### 2.2. Personal Variables

The personal variables were coded on the basis of socio-demographic data (gender, age, nationality, educational achievements) and five questions focusing on summative symptoms (e.g., “I often lack energy; I have stomach, back or head pain; I feel tired; I have difficulty sleeping; I feel fear or anxiety”).

### 2.3. School Burnout

School burnout was measured using the School Burnout Inventory (SBI) by Salmela-Aro, Kiuru, et al. [3] translated and validated in French by Meylan et al. [7]. Cronbach’s alpha for the French SBI scale is very satisfactory (0.82), as is Jöreskog’s rho (0.91). The SBI comprises 9 items, which assess three dimensions of burnout, emotional exhaustion, cynicism, and feelings of inadequacy. Emotional exhaustion is assessed by four items (e.g., “I feel overwhelmed by my school work”), cynicism by three items (e.g., “I continually wonder if my school work has any meaning”), and feelings of inadequacy by two items (e.g., “I often feel inadequate in my school work”). Each item is scored on a 6-point Likert scale, ranging from 1 to 6 (1 = completely false; 6 = completely true) with a possible total score of 54. The sum of all items provides a total score indicating the severity of the pupils’ burnout level: an overall score below 29 represents a low level, a score between 30 and 34 represents an average level, and a score above 34 represents a high level of burnout [7].

### 2.4. School Stress

School-related stress was measured by the Echelle des tracas quotidiens [38]. Inspired by the Adolescent Hassles Inventory [39] this scale was designed, translated, and validated in French by Plancherel et al. [38]. In the present study, as in Lacombe and Squillaci’s study [9], only the 25 items related to the school context were retained, in accordance with the objectives of the study. The items ask about relational difficulties with peers, with teachers or with parents on themes related to school, work, and academic and professional future. The scale measures both the frequency and intensity of daily worries of children and teenagers. For each item, pupils mention whether they have encountered difficulties (or not) in the past 6 months and their rating of how disturbed (hindered) they were. Each item is scored on a 5-point Likert scale from 0-4 (0 = I have not had this problem; 1 = I have had this problem, but it has not bothered me; 4 = I have had this problem and it has bothered me a lot) [38]. In this study, the SBI scale obtained a Cronbach’s alpha of 0.89, which represents a strong degree of internal consistency. The sub-dimensions reported a Cronbach’s alpha of 0.75 for emotional exhaustion, 0.80 for cynicism, and 0.73 for inadequacy. The daily hassles scale scored 0.89.

### 2.5. Interpersonal Variables

Interpersonal factors were assessed through questions on perceived support provided from friends, teachers, peers, and parents (e.g., “Do you feel supported by your friends?”). All of these items were scored on a 6-point Likert scale, from 1 to 6 (1 = completely false; 6 = completely true). The perception of the class climate was also collected by the question “do you like your class” (yes/no).

### 2.6. Organizational Variables

The organizational variables that were collected concerned the number of pupils per class and the number of teachers working in the class during a week.

### 2.7. Pandemic-Related Variables

Five factors were used to assess students’ experiences during the COVID-19 lockdown. Firstly, two questions asked about satisfaction with the supports in place “now” and during the lockdown (“I think the academic supports in place now to catch up on school material are sufficient”; “I think the academic supports in place for organizing distance learning during the lockdown were sufficient”). Secondly, three questions were asked about the perceived return to normality (“I think I have returned to my life as it was before the COVID-19: leisure activities, friendship, socializing”), about the perceived workload (“My workload has increased because of the health crisis”), and about the confidence in the future. All these items were scored on a 6-point Likert scale, from 1 to 6 (1 = completely false; 6 = completely true).

### 2.8. Statistical Analysis

The significance level was set at 0.05. Bonferroni correction was applied to all post-hoc tests to correct for cumulative error. The effect sizes reported correspond to Cohen’s d. To answer the first research question (prevalence of burnout), students were classified according to the degree of burnout they felt (low, medium, high). A score below 29.99 corresponds to a low level of burnout, a score between 30 and 34 corresponds to a medium level, and a score above 34 (≥34.01) corresponds to a high level of burnout [7]. To answer the second research question (comparison of mean scores between T1 and T2), *t*-tests were performed. Finally, for the third research question (risk and protective factors), *t*-tests for mean scores related to categorical variables (such as gender) and correlation analyses (for metric variables) were first performed to analyze single effects and then all significant variables related to personal, interindividual, organizational, and pandemic-related factors were integrated into four separate general linear models in order to combine categorical and metric variables and to analyze the predictive factors of school burnout. Descriptive analyses were performed in Excel and statistical analyses in SPSS 26.

## 3. Results

The results of the 2nd measurement time (T2) show a proportion of students with severe burnout in the sample of 11.4% (*N* = 21) out of the total sample (*N* = 184). Table 2 provides a comparison of the burnout levels of the samples at T1 (2014; *N* = 313) [9] and T2 (2022; *N* = 184). The Chi^2^ test indicates that the prevalences are significantly different between the two measurement times (x^2^ = 11.347 (2) *p* = 0.003)

To compare the mean scores of burnout and its subdimensions over the two measurement times, *t*-tests were performed. Table 3 indicates that the overall burnout score and its subdimensions in 2022 are significantly higher than in 2014. It is however worth noting that this score reflects a low level of burnout in the whole sample according to the cut-off points established by [7].

### Burnout Factors

Within individual factors, gender, somatic symptoms, academic performance, and perceived stress were analyzed in relation to the three dimensions of burnout. The results show that the mean scores between girls and boys do not vary significantly, either in global burnout (Boys M = 2.31; Girls M = 2.34) or in any of the three dimensions of the model.

Then, correlations were conducted to analyze the links between school burnout and the 4 categories of risk factors (individual, interpersonal, organizational, and COVID-19 related). Correlations between burnout and individual risk factors are all statistically significant (Appendix A). Regarding the somatic symptoms, results show the more the pupils report symptoms, the more they present high burnout levels. More specifically, lack of energy is more highly related to total burnout and cynicism (521 **). Fatigue is more strongly related to burnout and to emotional exhaustion (0.554 **). Regarding academic performances (in mathematics and in French), the correlations results are all negative and significant. These findings highlight links between high academic performance and low level of burnout. It is to be noted that the effect sizes of the correlations in French performance (−0.273 **) are smaller than those in mathematics (−0.460 **). Finally, correlations between students’ overall perceived stress and the level of burnout were analyzed. Several forms of stress have been identified through the Daily Worry Scale. All correlations were significant and positive, suggesting that perceived stress levels are related to burnout and its subdimensions. Success stress (“pressure to perform”) was the most strongly correlated with burnout (0.683 **), emotional exhaustion (0.657 **), and feelings of inadequacy (0.639 **). Relational stress was strongly related to emotional exhaustion (0.529 **), underlining the relevance of relationships for the pupils’ emotional well-being.

Of the interpersonal factors, support from teachers, classmates, friends, and parents were assessed. All the correlations were found to be statistically significant (Appendix A). Correlations are negative, meaning that the higher the perception of support, the lower the levels of burnout, both in total burnout and in the sub-dimensions. The strongest correlations are those of classmates and then of teachers. Finally, the classroom climate, measured with the item “Do you like your classroom” was subjected to a *t*-test. Results show statistically significant differences (*t*(182) = −4.194, *p* < 0.001) in burnout means between pupils who declare liking their class (*M = 2*.22, *SD =* 1.00) and those who do not (*M =* 3.22, *SD =* 0.99). Pupils who declare not appreciating their classroom climate scored significantly higher means for emotional exhaustion (*M =* 2.84, *SD =* 1.26; *t*(182) = −3.588, *p* < 0.001), for cynicism (*M = 3*.37, SD = 1.15; *t*(182) = −3.880, *p* < 0.001) and for inadequacy *(M =* 3.75, *SD =* 1.20; *t*(182) = −3.689, *p* < 0.001) than those who declare that they enjoy it (*M =* 2.02, *SD =* 0.93; *M =* 2.28, *SD =* 1.19; *M =* 2.54, *SD =* 1.40, respectively).

At the level of organizational factors, two variables related to the school context were tested: the number of students per class and the number of teachers per class. However, the analysis of variance showed a non-significant effect of the number of teachers on burnout (F(3,180) = 0.82, *p* = 0.484) and its dimensions, namely emotional exhaustion (F(3,180) = 1.21, *p* = 0.308), cynicism (F(3,180) = 0.14, *p* = 0.936), and inadequacy (F(3,180) = 1.05, *p* = 0.373). Regarding the number of students per class, the ANOVA results were also non-significant. The number of students did not influence burnout (F(3,180) = 1.16, *p* = 0.326) as well as feelings of emotional exhaustion (F(3,180) = 0.72, *p* = 0.539), cynicism (F(3,180) = 0.97, *p* = 0.407) and inadequacy of the students in this sample (F(3,180) = 1.59, *p* = 0.193). The average tables are available (Appendix A).

School burnout means were compared with the five items related to the COVID-19. Findings indicate that perceived increased workload is positively correlated with school burnout (0.402 **). Being confident about the future (−0.551 **) and feeling a sense of having regained their pre-pandemic life (−0.495 **) are the factors most significantly negatively related to burnout. This suggests the more positively future confidence is rated, the more positive pupils report having regained their pre-COVID life, the lower burnout is expressed. Adequacy of current supports was significantly and negatively correlated with burnout, emotional exhaustion, cynicism, and inadequacy.

All the significant variables were fitted to a general linear model to assess the predictive value of the overall pupil burnout (Table 4). 

A general significant model was found for the personal factors (*B* = 2.363 (*SE* = 0.432), *t* = 5.472, *p* < 0.001). In terms of somatic functioning, pain, fatigue, sleep difficulties, and fear were found to be predictive of school burnout. Among somatic symptoms, fear, and anxiety are the most predictive of burnout, but the effect size remains medium and is small for other somatic symptoms. In terms of school results, poor performance in mathematics predicted student burnout. Effect size is important for performance in mathematics. It is interesting to observe that the French scores are not predictive of the level of burnout here although the correlation was found to be significant. Finally, perceived high stress is the strongest predictor of burnout, which is consistent with the definition of burnout itself, since it is stress that exceeds the individual’s resources.

A general significant model emerged between school burnout and interpersonal factors (*B* = 6.127 (*SE* = 0.378), *t* = 16.225, *p* < 0.001). Findings indicate (Table 4) that the perceived supports from teachers, classmates, and parents are negative predictive factors for burnout acting as protective factors. In this sample, parent and classmate support had the greatest impact. Teacher support is also a protective factor, but the effect size is medium in this sample. Finally, as shown in Table 4, results highlight that a positive classroom climate perceived is a negative predictor of burnout, which means that pupils with a positive classroom climate are less likely to experience burnout than their peers

A general significant model emerged between school burnout and COVID-19 related factors (*B* = 5.082 (SE = 0.406), *t* = 12.503, *p* < 0.001). More specifically, findings (Table 4) indicate that feeling that one’s daily life is back to normal, the adequacy of existing supports and the confidence about the future were found to be negative predictive factors for burnout, whereas feeling an increased workload caused by the COVID-19 was found to be predictive of school burnout. It is interesting to note that the two most protective factors against burnout are the feeling that one’s daily life is back to normal. According to the effect size, this factor would be as strong as the mathematics results. In addition, having confidence in the future appears to be the second most important protective factor of all.

## 4. Discussion

Burnout is a syndrome that affects not only people in many careers, but also school-aged children. Research has shown that pupils can suffer from burnout most likely due to personal and interpersonal factors. While many studies have aimed at identifying and better understanding the particularly high risk for pupils, the present study is the first to compare the risk of burnout for preteens at two different measurement times and attempt to identify risk factors related also to the consequences of the COVID-19.

### 4.1. Burnout Prevalence

In the sample, 11.4% of students mentioned high levels of burnout (*M =* 12.38, *SD =* 0.48). In conformity with our hypothesis, the prevalence of burnout has significantly increased between 2015 and 2022. This result is in line with the Swiss national report [40], which found that 88.5% of children aged 11 to 15 considered themselves to be in good psychological health and that the prevalence of young people with depressive symptoms has increased. This result is slightly lower than those in the literature. The main hypothesis is that of age, as studies show an increase in burnout in the adolescent period. Indeed, as noted by Gabola et al. [8] and confirmed by numerous studies [37,41,42]. The older the students are, the more likely they are to experience high levels of burnout.

With regard to the overall burnout score (*M* = 20.96; *SD =* 9.43) measured in the sample, it is worth noting that this score corresponds to a low level of burnout in the whole sample according to the thresholds established by Meylan, Doudin, Antonietti et al. [7]. These data are consistent with the results of the literature which also find low overall burnout scores if the whole sample is taken into consideration [6,9,43,44,45]. The hypothesis is therefore validated. This low burnout rate in the overall sample can be related to the number of young people who have a positive attitude towards school. The Swiss national report [40] shows that 76% of young people like school. Comparing the overall scores measured in Switzerland among 8th grades pupils in 2014 and those measured in 2022, a significant increase is observed. This may suggest an effect in the mid-term of COVID-19 on the perceived health of preteen students at school. However, other factors could explain this difference. The latest national report on the health of children and adolescents in Switzerland [40] shows that the proportion of young people aged 11 to 25 suffering from psycho-emotional disorders (sadness, nervousness, anxiety, difficulty in falling asleep, depression) has increased over the last 10 years. The report also notes that excessive use of digital media can lead to feelings of loneliness and depression. Therefore, student burnout will be measured again in a similar sample in 2023, 2024, and 2025 as a follow-up to this research.

### 4.2. Burnout Factors

With regard to individual variables, the results do not show a gender difference. In the literature, only few studies report a higher level of burnout in girls [11,13], the most do not find a significant difference by gender [8,9,12,20,21,42]. Findings of this research are therefore consistent with the trend observed in the literature and validate our hypothesis. In terms of somatic symptoms, lack of energy seems to be related to cynicism, which corroborates the findings of the literature since cynicism is characterized by a loss of interest and motivation for school [23] Fatigue and anxiety are more strongly linked to the feeling of emotional exhaustion, which is precisely characterized by fatigue linked to the perception of being overworked [5]. The perception of being inadequate as a student (inadequacy) is strongly correlated in this study with the feeling of anxiety. In this respect, pain, fatigue, sleep difficulties, and fear appear to be predictive factors of school burnout for the students.

The impact of school performance, especially in mathematics, on burnout is high, since low performance is a predictor of burnout. These results confirm data from the literature and validate our hypothesis. For Virtanen et al. [12], academic performance is positively associated with cognitive and behavioral engagement. According to these authors, students with good results are more engaged, value school more, and use their resources better. Moreover, it is interesting to note that in our sample, inadequacy is the burnout dimension most strongly correlated with academic performance, and this is both in mathematics and in French. As mentioned by Maslach et al. [46], inadequacy is the self-reported dimension of burnout. These results suggest that a low-achieving child appears to perceive him/herself as less academically able and less accomplished than a pupil with high perceived achievement. In this regard, several authors [12,47,48] highlight the protective effect of high self-esteem and a high sense of self-efficacy on burnout. The results also show that school stress is a predictive factor for burnout, which is consistent with findings from the literature that define burnout as “a chronic stress syndrome” [23] (p. 33).

Among the interpersonal variables considered in the study, different sources of support were analyzed to understand their effects on burnout. All sources of support—parents, teachers, friends, peers—were found to be significantly and negatively correlated with burnout and its dimensions. These results validate our hypothesis. It is worth noting that only perceived lack of support from parents, peers, and teachers predicted school burnout. The protective factor with the largest effect size is parental support. This result confirms Sim’s [49] finding that the most important social support was that of the parents. For Zakary and Bendahman [50] the more parents take an interest in school work and trust their child, the more successful the pupils are. Conversely, lack of parent supports become a risk when parents devaluate their child or place high pressure on academic performance [51]. The study by Meylan et al. [6] show that parental support negatively predicts burnout and more specifically the level of cynicism and inadequacy. In our study, 85% of children say they are strongly supported by their family, which is close to the 90% reported by the Swiss national report on child and adolescent health [40]. In terms of support received in the school setting, the literature confirms that the more support students perceive from their teachers, the lower their levels of burnout [1]. According to Salmela-Aro et al. [1] teacher motivation, encouragement, and interest is negatively related to students’ burnout scores. Conversely, teacher psychological control increases student emotional exhaustion and cynicism [52]. In the research by Meylan et al. [6], the most important effect of perceived lack of support also concerns cynicism. It can be hypothesized that when support is perceived negatively, students become disengaged and feel more cynical towards school. In our study, 69.6% of students report a high degree of support from their teacher. This result can be put into perspective with the Swiss national report of 2020 [40]. In the latter, the majority of 11–13-year-olds (70%) say they have confidence in their teachers, while at 14–15 years old 57% of boys and 52% of girls have this opinion. This result may also explain the increase in burnout among adolescents compared to 11–13-year-olds.

The results indicate that peer support appears to be a protective factor against burnout in all three dimensions of the model. According to Estell and Perdue [53], peer support is thought to foster attachment and engagement in school, feelings that protect against burnout and its consequent symptoms. The results of the literature also point to the importance of support from peers who are positively engaged with school [12].

Finally, our results indicate that the assessment of the classroom climate influences the burnout averages, both in terms of emotional exhaustion, cynicism and inadequacy. These results corroborate those of the literature in particular the results obtained in our previous study conducted in 2014 [9] and tend to confirm our hypothesis. While a positively perceived classroom climate seems to protect students from burnout, a negative atmosphere can have the opposite effects [1] such as rumors spreading, a recognized risk factor in research [54].

With regard to organizational factors, the two factors investigated in this study were the number of pupils per class and the number of teachers per week. The results indicate that class size does not significantly affect the means of burnout, emotional exhaustion, cynicism, and inadequacy. Although class size reduction can improve academic performance [55], no study seems to have linked this factor to the problem of school burnout. As for the number of teachers worked with per week, this variable also does not lead to a significant difference between the averages for burnout, emotional exhaustion, cynicism and inadequacy. These results suggest that school burnout and its constituent symptoms are not influenced by quantity, but rather by the quality of interpersonal relationships.

Finally, the variables related to the COVID-19 pandemic show significant correlations with burnout, allowing the identification of one risk factor (workload) and four protective factors (getting back to life before COVID-19; adequacy of current support; adequacy of support during containment; trust in future). In view of the significant increase in the prevalence of burnout among students, these factors provide some insights into how the COVID-19 pandemic appears to have impacted the perceived health in the sample. The feeling of increased workload due to the COVID-19 was found to be a predictor of student burnout. These results are consistent with those of the study of Meylan and Hascoët [56] who report a more specific effect on emotional exhaustion. Indeed, according to Teuber et al. [57], Núnez-Regueiro [58], and Demerouti et al. [43], a high perception of demands is a risk factor for emotional exhaustion and student well-being. Meylan and Hascoët [56] find, among other variables, that workload is related to stress, with stress itself being associated with burnout and its three dimensions.

In response to perceived workload, the adequacy of faculty supports was assessed and the effect on burnout analyzed. Sufficiency of perceived current supports was found to be a negative predictor of burnout. As noted by Salmela-Aro and Upadyaya [59], the perception of sufficient and adequate resources enables the student to engage academically and manage perceived demands effectively. Finally, confidence in the future is a negative predictor of burnout. This variable therefore seems to act as a protective factor against burnout. These results corroborate those of Martos Martínez et al. [47], according to which future prospects and burnout are negatively correlated. According to these authors, students who perceive the future less positively experience more exhaustion, cynicism, and inadequacy than those who perceive it positively.

## 5. Conclusions

This research measured student burnout two years after the pandemic lockdown and identify risk and protective factors that could explain this syndrome. “The real measure of a country is the attention it gives to its children, their health and safety, their material, their health and safety, their material situation, their education and socialization, and their sense of being loved, valued and to be loved, valued and included in the families and societies into which they are born” [60] (p. 1). This quote alone shows the importance of regularly measuring school burnout in the same population in order to understand its evolution and to propose effective interventions. Results show a significantly higher average burnout than that measured in 2014 among the students surveyed, suggesting a possible effect of the COVID-19 period on students’ well-being in the medium term. Several factors were found to be predictive of burnout. At the individual level, perceived stress, low academic performance in mathematics and somatic symptoms such as lack of energy, fatigue, pain, sleep difficulties, and anxiety predict the onset of school burnout. At the interpersonal level, support from classmates, teachers, and parents are related to a decrease in burnout and summative symptoms and a negative classroom climate predicts student burnout. At the organizational level, no factor was found to be significant. Finally, with regard to factors related to the COVID-19 pandemic, students who indicated that they had returned to their pre-COVID-19 life, had confidence in the future, and received sufficient support to make up for their shortcomings were less likely to suffer burnout than their peers. These three factors are negative predictors of burnout. Conversely, an increase in workload due to COVID-19 is a predictor of burnout.

At the end of this research, several limitations should be mentioned. The first limitation is the type of questionnaire used. Indeed, these are self-report tools filled in by the students about themselves. There is therefore a risk of subjectivity in the answers. In addition, the variables measured in relation to the COVID-19 pandemic do not come from a standardized questionnaire, so further research would be necessary. Notwithstanding these limitations, the strength of the study is in having measured the means of two samples with comparable characteristics (age, education level, district, school environment, etc.), in order to provide a comparison on the burnout of two group of students. Another strength was to compare these scores once the pandemic was over, in order to analyze whether perceived health had returned to pre-pandemic normality two years later and, if not, to identify factors related to the syndrome. Finally, our study also points to some factors that would be interesting to study in the future, such as the level of teacher burnout in which the pupils are, or the impact of the use of digital media at home, which have been shown to be related to sleep difficulties and feelings of loneliness in previous studies [40].

### Educational Implications

Our results highlight some important points. Firstly, school burnout should not be neglected. It already appears in young students [9,16,61] and tends to increase over the years of study [41,48]. Thus, it is important to study young populations, in order to continue the knowledge of the phenomenon and specially to identify the first signs of emotional exhaustion, cynicism, and inadequacy in these students so as to be able to act as soon as the first symptoms appear to limit their consequences. Secondly, it is important to regularly measure the prevalence and scores of burnout in similar samples. Indeed, if this study has highlighted an increase in school burnout among preteen students between 2014 and 2022, repeated measurements make it possible to monitor the evolution of mental health reported by students, especially during periods of health, economic, or social crisis. Thirdly, this study highlights the large number of explanatory factors of school burnout. While the study of risk factors is important, the study of protective factors is equally important, particularly in post-crisis periods [42], so that supports can be adjusted by putting in place services that meet the needs of students in order to promote their development even in crisis situations [62]. This study also opens up perspectives for intervention. As mentioned, the factors with the greatest impact are: confidence in the future, perceived stress, parental support, and mathematics results. It is therefore possible to imagine an intervention targeting these four factors or specifically one of them. For example, it would be interesting to involve parents in schooling by offering open classes where parents could see their child in the school setting. Another possibility would be an intervention targeting mathematical ability. In this regard, many literature reviews highlight evidence-based interventions to improve mathematics achievement [63,64,65]. In terms of stress, it would be possible to teach coping strategies and to regularly measure students’ perceived stress and the causes of it. This would make it possible to intervene directly before the onset of school burnout. Finally, in terms of confidence in the future, it would be interesting to ask students about their desires for the future in order to set up projects that would motivate them (for example, by proposing internships in different professions).

## Figures and Tables

**Table 1 children-10-00823-t001:** Prevalence of school burnout.

First Author	Country	*N* (% Girls), M_âge_	Scale	Prevalence
Salmela-Aro (2012) [10]	FIN	687 (47.6%) *M* = 15, T1	SBI	8.6%
	FIN	749 (49.13%) *M* = 17.5, T4	SBI	12.5%
Lacombe (2015) [9]	SUI	313 (50.48%) Age: 11–13	SBI	4.5%
Kinnunen (2016) [11]	EUR	10,325 (52.4%) *M* = 15.19	SSBI	18.1%
Virtanen (2016) [12]	FIN	2485 (52.1%) *M* = 14.71	BBI-10	5.5%
Gerber (2018) [13]	SUI	249 (64.26%) ^1^ *M* = 16.09	SMBM	12%
	SUI	144 (32.64%) ^2^ *M* = 16.22	SMBM	6.9%
Lee (2019) [14]	KOR	1015 (57.3%) Age: 17–19	KABI	15.47%
Cheung (2019) [15]	CHN	1209 (39.8%) *M* = 14.85	MBI-SS	32.2%
Gabola (2021) [8]	SUI	343 (49.27%) *M* = 14.81, S1	SBI	14.6%
	ITA	497 (48.29%) *M* = 15.09, S2	SBI	27.6%
Summary	*K* = 11	*N* = 18,016		14.36%

Note. SUI = Switzerland; ITA = Italy; FIN = Finland; EUR = Europe, CHN = China; KOR = Korea. ^1^ Academic track; ^2^ Vocational track.

**Table 2 children-10-00823-t002:** Prevalence of burnout at T1 and T2.

Level of Burnout	Prevalence T1 (2014) *N* = 313	Prevalence T2 (2022) *N* = 184
low	91.4% *	82.6% *
medium	4.8% *	6.0% *
severe	3.8% *	11.4% *
Total	100%	100%

Note. T1 = Measurement time 1; T2 = Measurement time 2; * *p* < 0.05

**Table 3 children-10-00823-t003:** Descriptive data and *t*-test results in the three dimensions of burnout.

	T1 (*N =* 313)		T2 (*N =* 184)			
	*M*	*SD*	*M*	*SD*	t (…)	*d*
Burnout	1.97	0.85	2.33	1.05	3.929 ***	0.365
EE	1.87	0.84	2.11	1.00	2.692 **	0.25
CYN	1.91	1.08	2.39	1.23	4.465 ***	0.415
INAD	2.28	1.28	2.67	1.43	3.111 **	0.289

Note. ** *p* < 0.01; *** *p* < 0.001; EE = Exhaustion; CYN = Cynicism; INAD = Inadequacy.

**Table 4 children-10-00823-t004:** Results of the general linear model between the different risk factors and burnout.

	*B*	*SE*	*t*	*p*	Eta-Square
Personal factors					
Somatic symptoms					
Lack of energy	0.048	0.049	990	0.323	-
Pain (stomach, head, back)	0.080	0.033	2.422	0.016	0.033
Fatigue	0.101	0.045	2.227	0.027	0.028
Sleep difficulties	0.067	0.033	2.018	0.045	0.023
Fear, anxiety	0.126	0.048	2.744	0.007	0.042
School results					
Mathematics average	−0.386	0.081	−4.771	<0.001	0.117
French average	0.038	0.080	0.470	0.639	-
Perceived stress					
Total perceived stress	0.684	0.113	6.056	<0.001	0.177
Interpersonal factors					
Support from teachers	−0.135	0.057	−2.345	0.020	0.030
Support from parents	−0.290	0.059	−4.923	<0.001	0.119
Support from friends	−0.110	0.062	−1.767	0.079	-
Support from peers	−0.238	0.056	−4.235	<0.001	0.091
Classroom climate	−0.811	0.219	−3.698	<0.001	0.071
Pandemic factors					
Increased workload since COVID-19	0.174	0.054	3.217	0.002	0.055
Back to life as before COVID-19	−0.231	0.048	−4.779	<0.001	0.114
Adequacy of current support	−0.124	0.055	−2.271	0.024	0.028
Sufficiency of support during containment	−0.013	0.045	−0.282	0.779	-
Trust in the future	−0.378	0.065	−5.830	<0.001	0.160

## Data Availability

The data presented in this study are available on request from the corresponding author. The data are not publicly available due to the fact that they are children and that the research indicated that the data would only be stored on a secure server at the university of Fribourg.

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
