# Peer review of "School Burnout after COVID-19, Prevalence and Role of Different Risk and Protective Factors in Preteen Students"

_children, 2023, doi:10.3390/children10050823_

Round 1
Reviewer 1 Report
This study compares two materials; one was collected in 2014, and the other later. The study looks at three areas of individual, interpersonal, and pandemic-related variables. When you look at burnout, you measure how much the individual estimated his burnout on a scale that he then divided into three groups, low, medium, and severe. This was then tried to predict based on individual, not personal and organizational factors. The study showed that the material from 2022 had almost three times the percentage of students who estimated burnout. That there was no difference between boys and girls in the later material and that those in need of support estimated higher burnout. Some psychosomatic complaints were more related to burnout than others in regression. When it came to school results, it was particularly maths rather than French. Support, especially from parents and peers, followed by the teacher, reduces the risk of burnout, and the individuals who did not appreciate their classroom climate had higher emotional exhaustion. When it came to the covid-related variables, an increased workload increased burnout, while the return to life as before, more support, and a great future reduced the risk.
What is the overall feeling of the study? 2014 was a long time ago, and more things than covid have happened since then; it is not clear if you control this; digitization here is one such thing that has happened in these years. However, I agree that covid has played a big role. The result of this study indicates that supporting students reduce the impact of mental health problems, specifically burnout in this study. Looking at mental health problems, there are many studies; the novelty is that they have some before and after measures of burnout.
While I like the division into individual, interpersonal, and organizational, I am a bit skeptical about school climate as a measure; many studies that look at school climate use the measures of peer and teacher support as well as aspects such as perceived bullying, which is, things that are clearly interpersonal if not specific aspects located higher up in the structure. Just liking or perceiving support is not enough to be organizational. I don't think it is clear here how your school climate differs from the interpersonal factor. As the question "Do you like your classroom" appears first in the result, it is not presented in the method, and I do not see this as an organizational variable but rather a measure of interpersonal or personal level.
The question concerning the support measure (see table 4) can perhaps rather be seen as organizational, but it is not clearly described in the method. It is also reflected quite a bit in the discussion, and I am not so sure that things like spreading rumors are actually organizational; it is definitely the school climate, but based on the support that is available from peers and teachers, some of this is interpersonal. Overall, the study describes fairly expected results, although, of course, focusing on what matters, such as returning to routines, having faith in the future, and providing sufficient support, is central.
There are a lot of tables, and it feels like they wanted even more tables, but the tables are not efficient; they could be condensed and thus remove information that is now in running text. Although I like table 1 that it gives a quick overview, this could perhaps be shortened or made more clear. It takes up different scales, but I can't see that this is problematized in the text; either that information is important, and then it is explained why, or it is not important, and then it is removed.
When it comes to the title, I do not really feel that "what about risk and protective factors?" What about them? You are looking at specific factors, and I do not feel that you can answer the title, and perhaps it needs to be closer to your aim.
Introduction: table 1: what does END stand for? If we look at table one, it is also important to think that these different studies have different age ranges. Mental health problems, in general, has a clear gradient with age; that is to say, the studies show that 15 years or older will measure higher levels than at 11 to 13. this is not something that you problematize.
Method:
You do not explain the 2014 group at all, how the data was collected, and what the ethics look like there. Short procedure. Not clear informed consent from parents was collected. 11-13 years. If it is not needed, this needs to be clarified why.
School burnout. What are the max and min of the composite scale? It is not made clear if my 9 or 0. It affects how you look at the cutoff of 34.1. this scale is only said to be validated but not in what way. Why not an alpha here that you have for school stress?
2.5 it is quite comprehensively formulated. If all questions were formulated in the same way, where would the question concerning the need for special support come from?
2.6 here, you bring up two-dimensional covid related factors, but I don't see that you actually use it because here you have organizational factors.
As for analysis, I don't understand why you don't have any background variables you check for; gender and age, which you mention anyway, are relevant.
In table 2, you could tell us a little about the cutoff levels because it is not completely clear in the method
In table 3, you mention the abbreviations, but not in table 4. which one seems to fit in the lines?
There is a lot of discussion on page 7 about material that is not in table 5. I think this part can be shortened; if it is in the supplement, it is because it was not important, then you should not bring up every correlation afterward. Then it was chosen that it is a supplement. The space the explanation of table 5 gets is too short in relation to what the explanation of a table in the supplement gets. Because even though all the variables that were relevant were entered into the linear model, not all came out, and it is more interesting to discuss. The text under the table really only says what is on the table, nothing more.
When we talk about interpersonal variables, this talks about support; here, 3 play a role, and central seems to be support from parents, followed by support from classmates, and finally, teachers. Eta-square tells more than just a significance, but that doesn't feel like it has been considered. Neither has aspects such as gender and age taken into consideration despite the fact that these variables usually interact with perceptions of support.
Yes, I already addressed the discussion about the organizational factors, but I don't think "do you like your classroom" or positive classroom climate. Means and sd could have been at the table if you're still talking about those who like their classroom and those who don't. However, it is not clear how this division took place.
Discussion:
Even in the discussion, average values are standard deviations; the experience becomes a discussion that is not raised; it is too close to what I have already read. To then say that this is the line between what was expected and previous research, what is then new?
Saying that it is good to be high-functioning and have high self-esteem, when you discuss on p. 10, needs to be put in relation to the fact that the group of children in need of support and especially mathematics, was affected, which was also seen in previous research, but not really highlighted here. Rather than saying school stress is a risk factor. Since you see it as surprising, it makes me wonder what kind of question you asked. Because I interpret this as the group of children who are in need of special support and that group has experienced special challenges. But if this is something else, it needs to be made clear through the text.
Author Response
What is the overall feeling of the study? 2014 was a long time ago, and more has happened than covid since then; it's not easy to control that; digitization is one of the things that has happened in recent years. However, I agree that covid has played an important role. The results of this study indicate that student support reduces the impact of mental health issues, particularly burnout in this study. In terms of mental health issues, there are many studies; the novelty is that they have measures before and after burnout.
- 10 L.645-650 Thank you for this comment. We included additional influencing factors as mentioned in the recent report on the health of children and adolescents in Switzerland (2020). A clarification has also been added concerning that the research would again be conducted in 2023, 2024 and 2025
While I appreciate the division into individual, interpersonal, and organizational, I am a bit skeptical of school climate as a measure of organizational factors; many studies that examine school climate use measures of peer and teacher support as well as aspects such as perceived bullying, i.e., things that are clearly interpersonal if not specific aspects higher up in the structure. Liking or perceiving support is not enough to be organizational. I don't think the difference between school climate and the interpersonal factor is clear. The question "Do you like your class?" appears first in the result, but it is not presented in the method, and I do not consider it an organizational variable but rather a measure of interpersonal or personal level. The question regarding the measure of support (see Table 4) can perhaps be considered an organizational variable, but it is not clearly described in the method. It's also reflected pretty broadly in the discussion, and I'm not sure that things like rumor spreading are really organizational; it's certainly school climate, but based on the support that is available from peers and teachers, some of that is interpersonal. Overall, the study describes fairly expected results, although of course it is critical to focus on what matters, such as getting back into the routine, having faith in the future, and having enough support.
- 3 L.150-161 We thank you for this relevant comment. Indeed, it is possible to consider school climate as an interpersonal variable, if we focus for example on peer relations (e.g. spreading rumours). We considered it as an organisational variable with respect to the influence of the climate by the number of pupils per class, the type of pupils, the number of teachers, the framework of intervention set by the teachers, etc. A clarification on this point has been added in the text (specify page)
There are a lot of tables, and it feels like they wanted even more tables, but the tables are not efficient; they could be condensed and thus remove information that is now in the running text.
- 8-9 In order to simplify the results, we have made a single table for the regressions
While I like Table 1, which gives a quick overview, it could perhaps be shortened or made clearer. It includes different scales, but I don't see how this is problematized in the text; either this information is important, and then it is explained why, or it is not important, and then it is removed.
- 2-3. L.84-130 We have provided more details on the different results regarding the prevalence of burnout and the scales used.
In terms of the title, I don't really feel like it says "what about risk and protective factors?". What about them? You're looking at specific factors and I don't think you can answer the title, which maybe should be closer to your goal.
Reformulation
School burnout after covid-19, a measure of the prevalence and role of different risk and protective factors in preteen students
- 1 L.2-3 We changed the title like this: School burnout after covid-19, prevalence and role of different risk and protective factors in preteen students
Introduction: Table 1: What does END mean? If we look at Table 1, it is also important to consider that these different studies have different age ranges. Mental health problems, in general, have a clear gradient with age; that is, studies show that 15 years or older measure higher levels than 11 to 13 years.
- 2 This is a mistake in the typo. It is the abbreviation for Finland (FIN), the change was done in the text
Method:
You don't explain the 2014 group at all, how the data was collected and what the ethics are. Abbreviated procedure. Informed parental consent was not clearly obtained. 11-13 years old. If not required, it should be stated why.
- 4 L.x230-245 Details have been provided and a reference to our 2015 article offers further details if required.
School burnout. What are the maximum and minimum values on the composite scale? It is not specified if it is a 9 or a 0. This affects how you view the 34.1 cutoff. this scale is only said to be validated but not in what way. Why not have an alpha like for school stress?
- 5 L.336 The scale has 9 items scored from 1 to 6 so the total is 6x9. We have added the total possible score (54) and Cronbach's Alpha for the SBI.
2.5, it is worded very comprehensively. If all the questions were phrased the same way, where would the question about needing special support come from?
- For Covid, the aim was to assess the specific support linked to the need to catch up certain content (setting up of private lessons, tutoring, etc.) and not to assess the general support perceived throughout the year by the pupils. This is why this detailed formulation has been clarified.
2.6 here you mention the two-dimensional factors related to covid, but I don't see that you actually use it because here you have organizational factors.
As for the analysis, I don't understand why you don't check any of the baseline variables; gender and age, which you mention anyway, are relevant.
- Thanks for this precision. We have verified all the personal, interpersonal and organisational variables present in the scales. We have not systematically reported the effects of all the variables, as our study (see title) focuses on the influencing factors. However, it is right to make a clarification in this respect, which we have done in the text.
In Table 2, you could tell us a little more about the thresholds, because it's not entirely clear in the method
- As we have produced a single table, the thresholds are specified more clearly.
In Table 3, you mention abbreviations, but not in Table 4. Which abbreviation seems to correspond to the lines?
There is a lot of discussion on page 7 about material that is not in Table 5. I think that part can be shortened; the reason it's in the supplement is because it wasn't important, and you don't have to go back to every correlation later. Then it was decided that it was a supplement. The space reserved for explaining Table 5 is too short compared to the space reserved for explaining a table in the supplement. Indeed, although all the relevant variables were entered into the linear model, not all of them emerged, and it is more interesting to discuss them. The text below the table actually says only what is on the table, nothing more.
- 7-8-9 L Clarifications have been made for the regressions and correlations. Some comments on the correlations have been maintained as they facilitate the understanding of the results
When we talk about interpersonal variables, we are talking about support; here, 3 play a role, and the main one seems to be parental support, followed by classmate support, and finally teachers. Eta square is more than just a meaning, but it does not seem to have been taken into account. Aspects such as gender and age were also not considered, even though these variables generally interact with perceptions of support.
- As no significant differences were found for gender and age, therefore, these were not included in the regressions
Yes, I've already addressed the discussion of organizational factors, but I don't think the question "do you like your class" or the positive classroom climate is taken into account. The mean and standard deviation could have been included in the table if we are still talking about those who like their class and those who do not. However, it is not clear how this division took place.
- Thank you, the change has been done.
Discussion:
Even in the discussion, the mean values are standard deviations; the experiment becomes a discussion that is not raised; it is too close to what I have already read. To then say that this is the dividing line between what was expected and previous research, what then is new?
Saying that it's good to be successful and have good self-esteem, when you talk about it on page 10, has to be related to the fact that the group of kids who needed support, especially in math, was affected, which has also been observed in previous research, but not really highlighted here. Rather than saying that school stress is a risk factor. Since you find that surprising, I wonder what kind of question you asked. Because I interpret that as the group of kids who need special support and have experienced special challenges. But if it's something else, you need to clarify that in the text.
10-11-12 I have added clarifications in the discussion to add things that are not in the results
Reviewer 2 Report
Thank you for allowing me to review the manuscript "School burnout after covid-19, what about risk and protective factors in preteen students?
The paper is well written and the point of view of student burnout, which is often overlooked, is interesting.
Some comments are listed below and I hope they will be helpful to the authors in revising the paper
- In the introduction, the authors define student burnout. The literature is up-to-date and well-researched. I would ask the authors to go into detail about the grade level of the educational system they are considering and to clarify already in the Introduction whether there are differences (e.g., does Burnout of middle school students have different characteristics than high school students? Does it also depend on the age group?). The authors state that "many studies measure Burnout in adolescents (13-18 years)." I would ask them to elaborate on this.
- The authors state "The present study aims to measure the burnout of 12-year-old students. Is this correct? I think it is more appropriate to say the grade level of the classes in which the data collection was carried out.
- The authors do not discuss the study context (a careful description of the Swiss education system would be desirable). Also, are there any studies on student burnout in this context to compare with the results of this study?
- I would ask the authors to detail the hypotheses and then discuss them in the discussions
- The authors present the spin-offs in educational terms. I would also ask for the possibility to address the fallout in terms of interventions.
- Finally, can teacher burnout affect student burnout? I would ask the authors to discuss this also in terms of future study perspectives.
Author Response
Thank you for allowing me to review the manuscript "School burnout after covid-19, what about risk and protective factors in preteen students?
The article is well written and the perspective of school burnout, which is often overlooked, is interesting.
Some comments are listed below and I hope they will be helpful to the authors in revising the article.
- In the introduction, the authors define student burnout. The literature is current and well documented. I would ask the authors to detail the grade level of the educational system they are considering and to clarify in the introduction if there are differences (e.g., does burnout in middle schoolers have different characteristics than in high schoolers? Does it also depend on the age group?). The authors state that "many studies measure burnout in adolescents (13-18 years old)." I would like them to elaborate on this point.
The school system has been explained in the method section in order to clarify the Swiss system. The results show an increase in the burnout rate during adolescence and differ mainly between countries and study paths. The demanding (academic) study paths increase the burnout rate. Otherwise the same risk factors are found in pre-teens and teens. I'm not sure if this is enough: it should be said that a clarification was made in the text.
- The authors state that "the purpose of the present study is to measure burnout in 12-year-old students." Is this accurate? I think it is more appropriate to state the grade level of the classes in which the data collection was conducted.
- 4 L.203 Thank you, the change has been done.
- The authors do not discuss the context of the study (a thorough description of the Swiss educational system would be desirable). Also, are there any studies of student burnout in this context that could be compared to the results of this study?
- 4 L.243-249 A description of this topic has been added to the methodology. There are at least three other studies in Switzerland, but they concern a public of adolescents (older than our sample), which makes it difficult to compare prevalence. However, the results were compared for risk factors. A reference to the Swiss national report on the health of children and adolescents was added in order to make comparisons with the same age and the same country.
- I would ask the authors to detail the assumptions and then discuss them in the discussions
- 4 L.212-230 Thank you, the change has been done.
- The authors present the impact in pedagogical terms. I would also ask for the opportunity to address the impact in terms of interventions.
- 13 L.933-945 Thank you, the change has been done.
- Finally, can teacher burnout affect student burnout? I ask the authors to discuss this also in terms of future study perspectives.
This perspective was added in conclusion
- 13 L.912-914 This perspective has been added in conclusion